# Lyophilized Extract from the Larvae of the Blowfly *Lucilia sericata* as a New Strategy for the Management of Chronic Wounds

**DOI:** 10.3390/biomedicines13030582

**Published:** 2025-02-26

**Authors:** Norman-Philipp Hoff, Falk Peer Gestmann, Theresa Maria Jansen, Sarah Janßen, Sabine Petersdorf, Bernhard Homey, Peter Arne Gerber, Heinz Mehlhorn

**Affiliations:** 1Department of Dermatology, University Clinic Duesseldorf, Heinrich Heine University, 40225 Duesseldorf, Germany; theresa.jansen@gmx.de (T.M.J.); sarah.janssen@med.uni-duesseldorf.de (S.J.); bernhard.homey@med.uni-duesseldorf.de (B.H.); prof.gerber@dermatologie-am-luegplatz.de (P.A.G.); 2Institute for Zoomorphology, Cell Biology and Parasitology, Heinrich Heine University, 40225 Duesseldorf, Germany; gestmann@alphabiocare.de (F.P.G.); mehlhorn@uni-duesseldorf.de (H.M.); 3Alpha-Biocare GmbH, 41468 Neuss, Germany; 4Institute for Clinical Chemistry, Transfusion and Laboratory Medicine, Berufsgenossenschaftliches Universitätsklinikum Bergmannsheil GmbH, Bürkle-de-la-Camp-Platz 1, 44879 Bochum, Germany; sabine.petersdorf@bergmannsheil.de

**Keywords:** chronic wounds, hard-to-heal wounds, maggot therapy, biological debridement, wound management, biofilm, infections, blowfly, *Lucilia sericata*

## Abstract

**Background/Objectives**: Chronic wounds represent a growing challenge for the aging population, significantly impairing their quality of life, increasing the frequency of medical consultations, and imposing substantial healthcare costs. Chronic wounds are prone to complications, including local and systemic infections, and in severe cases, amputations. The therapeutic use of live larvae from the blowfly *Lucilia sericata* (biological debridement) has regained attention for its ability to debride necrotic tissue and stimulate granulation. Despite its benefits, this therapy is constrained by the logistical challenges of producing and delivering live larvae and by patient adherence issues. **Objectives**: This study aimed to develop a lyophilized extract of *Lucilia sericata* larvae and evaluate its efficacy in treating chronic wounds. **Methods**: A lyophilized maggot extract (Larveel^®^, Alpha-Biocare GmbH, Neuss, Germany) of the larvae of *Lucilia sericata* was produced under GMP conditions. In a total of ten patients with chronic refractory wounds, the extract was used in individual therapeutic trials and its effect on bacterial colonization and wound healing was investigated. Wound healing was documented photographically and measured visually in terms of a reduction in the wound slough, an increase in the granulation tissue, and a reduction in the wound area. **Results**: Of the ten patients, three discontinued their treatment due to *P. aeruginosa* colonization. In seven patients, significant fibrin reduction, granulation, and wound healing occurred, with two achieving complete closure and four showing advanced epithelialization. **Conclusions**: In 7 of the 10 patients, the application of the extract led to a marked reduction in wound slough, and improved granulation and the progression of wound healing. These effects are likely attributable to the extract’s ability to disrupt bacterial biofilm formation. The findings suggest that this novel therapeutic approach may provide a practical and effective alternative to live larval therapy for managing chronic wounds.

## 1. Introduction

Chronic non-healing wounds represent an increasing challenge for the aging populations of industrialized nations, often resulting from multifactorial etiologies. Chronic wounds, typically defined as wounds that fail to demonstrate significant healing progress within three months, pose a substantial challenge to modern medicine. According to Körber et al., chronic wounds in the lower extremities frequently arise as complications of venous, arterial, or diabetic diseases [1]. In Germany, the prevalence of chronic wounds is estimated to be 2–3 million individuals, with leg ulcers affecting 0.7% of the general population and over 3.38% of individuals aged ≥80 years [2]. Chronic venous ulcers alone account for a substantial socio-economic burden, consuming approximately 2% of the national healthcare budget, with an average annual treatment cost of EUR 6600 per patient [3,4,5]. The effective management of chronic wounds is particularly challenging against the backdrop of increasing healthcare economization and cost containment. Optimal wound care necessitates a multifaceted approach that combines etiological treatments with appropriate local wound management. The intricate and multifaceted process of skin wound healing is fundamental for restoring tissue function and integrity. The skin’s efficient wound repair capability is crucial for maintaining bodily equilibrium and defending against external threats, including infections. As such, deciphering the underlying mechanisms of this process is of critical importance. This knowledge facilitates the development of powerful therapeutic interventions to promote wound healing and improve patient outcomes [6]. The selection of a therapy is guided by factors such as wound size, location, exudate levels, and microbial colonization. Moist wound healing, the current gold standard, aims to convert chronic wounds into acute wounds to facilitate granulation and re-epithelialization [7].

However, certain wounds remain unresponsive or develop critical bacterial colonization despite conventional therapies, necessitating alternative treatment modalities. The biological treatment of chronic wounds, particularly biosurgery utilizing live *Lucilia sericata* larvae, has garnered renewed attention. Initially described during the European wars of the 19th and 20th centuries, biosurgery declined in popularity with the advent of antibiotics, but was rediscovered in the 1980s due to increasing antibiotic resistance and the rising prevalence of chronic wounds. With the increasing prevalence of non-healing wounds and the growing adoption of maggot therapy globally, an increasing number of clinicians are recognizing the efficacy of maggot therapy [8].

Larval secretions and excretions have demonstrated the ability to debride necrotic tissue, eliminate bacteria, and stimulate granulation, rendering biosurgery a valuable therapeutic option [9]. Maggots possess the ability to promote healing by triggering molecular mechanisms in the affected area through their enzymatic secretions. The enzymes present in the excretions/secretions (ESs) of larvae can activate healing processes in a wound site. The research has demonstrated that the ESs exhibit various beneficial properties, including antibacterial, antifungal, anti-inflammatory, angiogenic, proliferative, hemostatic, and tissue-regenerating effects, both in living organisms and laboratory settings [10]. It is believed that these properties enhance and expedite wound healing by initiating direct signaling cascades within cells in the injured area. The effectiveness of this treatment is also partly attributed to the diverse array of antimicrobial peptides (AMPs) that these larvae release into a wound [11]. However, the specific AMPs within the ESs remain largely unidentified.

Despite its efficacy, biosurgery presents several limitations [12]. Patients frequently experience aversion or dysesthesia due to the visible movement of larvae on wounds [13]. The distinctive, unpleasant odor associated with larval excretions can impair compliance, while logistical challenges, such as the production, storage, and timely delivery of live larvae, further complicate its implementation. In numerous regions worldwide, maggot containment dressings (bagged maggots) are not available for wound debridement due to their increased complexity and higher production costs compared to confinement (’free-range’) maggot dressings [14]. Biosurgery typically serves as a preparatory step in wound management, necessitating a transition to conventional therapies for complete healing. Contraindications, such as coagulopathies, proximity to large vessels or body cavities, and *Pseudomonas aeruginosa* infections, limit its applicability [15].

Given these challenges, research has focused on isolating and examining the individual components of *Lucilia sericata* larvae for their therapeutic potential. Studies have demonstrated that the larval components exhibit beneficial effects on wound healing, suggesting their potential as an alternative to live larval therapy [16,17]. To date, these beneficial effects on wound healing have only been observed in ex vivo or animal studies. Considering the limitations of traditional maggot therapy and the current findings from ex vivo models, our objective was to develop a maggot extract and evaluate its efficacy on a cohort of patients no longer receiving treatment as part of a proof-of-principle study.

In contemporary clinical settings, the prevalence of patients with complex wounds is increasing, particularly among those who have undergone treatment or have chronic conditions. Concurrently, patients generally exhibit a preference for outpatient care to minimize extended hospital stays. Moreover, the healthcare industry, notably in Germany, faces escalating financial constraints. These factors support the development of a straightforward, cost-effective maggot therapy that can be administered on an outpatient basis and is comparable to other topical treatments, such as autolytic and enzymatic wound debridement.

The objective of this study was to test a sterile, storable larval extract with comparable efficacy to live larvae in its effectiveness and safety, but without the associated disadvantages. A larval extract can be applied as a conventional topical agent, minimizing odors, mitigating patient discomfort, and enabling treatment to continue until complete wound closure.

## 2. Materials and Methods

### 2.1. Development of a Lyophilized Maggot Extract

In accordance with the EU Good Manufacturing Practice (GMP) guidelines for human and veterinary medicinal products, a sterile lyophilized extract of *Lucilia sericata* larvae was developed (Larveel^®^, Alpha-Biocare GmbH, Düsseldorf, Germany) (Figure 1A). Third-instar larvae produced under GMP conditions for pharmaceutical excipients were utilized for extract preparation. The larvae were rapidly frozen in liquid nitrogen, homogenized in a knife mill with an equal amount of pharmaceutical-grade water to form a uniform emulsion, and subjected to thermal treatment at temperatures exceeding 60 °C. Following sedimentation and filtration, a sterile-filtered ultrafine filtrate was obtained. The filtrate (2 mL) was subsequently transferred into glass vials and lyophilized, followed by γ-irradiation for sterilization. Prior to application, the extract was reconstituted in 2 mL of physiological saline.

### 2.2. Application of Lyophilized Maggot Extract to Patients with Chronic Wounds

The extract was employed in individual therapeutic trials for chronic, therapy-resistant wounds following ethical approval by the Heinrich Heine University Düsseldorf Medical Faculty. The patients provided informed consent for treatment with lyophilized maggot extract, wound exudate sampling, and scientific data generation. Ten patients with chronic wounds were included based on the following criteria:Chronic leg ulcers;Refractory to treatment for at least three months;No underlying consumptive diseases (e.g., malignancies);A three-week pretreatment phase with stage-appropriate wound care demonstrated no significant improvement.

Following obtaining informed consent, the patients underwent the following protocol:Comprehensive clinical and wound-specific history;Standardized photo documentation;Wound exudate collection;Microbiological sampling;Application of *L. sericata* extract;Dressing with sterile wound coverings.

Exudate samples were collected prior to treatment initiation and at weekly intervals thereafter to analyze potential wound-healing mediators. For each 10 cm^2^ wound surface, one vial of extract reconstituted in 0.9% saline was applied. Following a five-minute drying period, sterile dressings comprising nonadhesive gauze, gauze compresses, and elastic wraps were applied. Dressing changes were performed every other day, either independently or with caregiver assistance. Weekly clinical evaluations, including photographic documentation and adverse effect monitoring (e.g., pain), were conducted at the interdisciplinary ulcer clinic of the Dermatology Department, University Hospital Düsseldorf. The treatment regimen was maintained for a maximum duration of eight weeks (Figure 1B). Fifty percent of the patients received treatment thrice weekly at the clinic. Four patients managed dressing changes twice weekly at home with a weekly clinic visit. One patient independently performed all dressing changes.

### 2.3. Patient Demographics and Wound Characteristics

This study’s cohort comprised 10 patients (mean age, 66 years for females and 74 years for males). Ulcer duration ranged from three months to >three years, with a conservative mean duration of 15 months (Table 1).

### 2.4. Exudate Collection from Ulcers

Following dressing removal, the wounds were irrigated with sterile saline and dried. Saline was applied for two minutes, then aspirated into syringes. Samples were centrifuged at 500× *g* for five minutes at 4 °C to remove cells and debris, filtered through 0.45 µm and 0.22 µm syringe filters, and stored in liquid nitrogen.

### 2.5. Bacterial Colonization Analysis

Swab samples were obtained with eSwab extraction systems (Copan Flock Technologies srl., Brescia, Italy) from the wound edges and center prior to treatment initiation and biweekly thereafter.

The swabs were automatically spread on Columbia agar with 5% sheep’s blood (bioMérieux Deutschland GmbH, Nürtingen, Germany) and MacConkey agar (Oxoid Deutschland GmbH, Wesel, Germany) and incubated for 48 h at 37 °C. Identification and resistance testing was carried out automatically using a VITEK II^®^ (bioMérieux Deutschland GmbH, Nürtingen, Germany) or by agar diffusion for pseudomonads. Resistance was assessed according to the CLSI (Clinical and Laboratory Standards Institute) standard. Bacterial identification and resistance testing were performed at the Institute of Medical Microbiology and Hospital Hygiene, Düsseldorf. The isolates were cultured for subsequent analysis. For further experiments, Columbia agar plates with 5% sheep’s blood were inoculated with 3 to 5 colonies of the permanent cultures of the respective isolates and incubated overnight at 37 °C. The cell count was adjusted spectrophotometrically to 0.5° McFarland (DensiCHEKTM plus, bioMérieux Deutschland GmbH, Nürtingen, Germany) in saline solution. The resulting bacterial suspension was used for further experiments within 20 min.

### 2.6. Effect on Bacterial Biofilms

For comparability with the results of other authors on the effect of components of *Lucilia sericata* on bacterial biofilms, for this lyophilized maggot extract the same method was used as described by van der Plas et al. and O’Toole and Kolter [18,19]. Biofilms were formed on microtiter plates containing *S. aureus* (MSSA and MRSA), *S. pyogenes*, and *P. mirabilis,* as clinically frequently relevant and less frequently wound-colonizing reference species, as well as the clinical isolates obtained as described in Section 2.5 from wounds of patients treated with the lyophilized maggot extract.

The lyophilized maggot extracts were dissolved in biofilm medium and piperacillin/tazobactam stock solution was diluted in biofilm medium. A total of 100 μL of these solutions or biofilm medium alone were placed in round-bottom microtiter plates (FalconTM, Corning Incorporated Life sciences, Acton, MA, USA). Tests were inoculated with 5 μL of the 0.5 McFarland overnight bacterial suspensions produced as described in Section 2.5 and incubated for 24 h at 37 °C. After this period, the medium was removed and the wells were carefully rinsed three times with 300 μL tap water. Subsequently, 125 μL of a 1% ethanolic crystal violet solution was added to each well and incubated for 15 min at room temperature. After removing the crystal violet solution, the wells were washed four more times with 300 μL of tap water. Crystal violet bound to the biofilms was eluted in 200 μL of 30% acetic acid per well and incubated for a further 15 min. The solutions were transferred to fresh flat-bottom microtiter plates and the absorbances were measured at a wavelength of 570 nm and a reference wavelength of 620 nm.

### 2.7. Effect on Bacterial Growth

Wells of untreated polystyrene round-bottom microtiter plates (Corning Incorporated Life sciences, Acton, MA, USA) containing 100 µL tryptic soy broth (TSB), lyophilized maggot extracts dissolved in TSB, or piperacillin/tazobactam stock solution diluted in TSB were inoculated with 5 µL of bacterial suspension of the bacterial isolates and reference strains as described in Section 2.5. Growth inhibition was assessed by measuring the optical density (600 nm) after 24 h of incubation at 37 °C.

## 3. Results

### 3.1. Isolated Bacterial Species and Effects of Maggot Extracts on Bacterial Colonization In Vivo

The wounds of all the patients were permanently or intermittently populated by various bacterial species (Figure 2). The differentiation of the isolates from the same species was based on antibiograms. The most commonly isolated bacterial species were *Pseudomonas aeruginosa* (7/10 wounds), *Staphylococcus aureus* (6/10 wounds), and *Proteus mirabilis* (2/10 wounds) (Table 2). All the *S. aureus* isolates were methicillin-sensitive (MSSA). The species *Achromobacter xylosoxidans*, *Enterococcus faecalis*, *Citrobacter koseri*, *Providencia rettgeri*, Group G streptococci, and *Klebsiella pneumoniae* were each isolated only from the wounds of one patient. In addition to these single and multiple findings, the smears included (in part temporal) other bacterial species that were germs of the normal dermal flora and therefore not used for further analysis and assessment. Essentially, a continuous decrease in the bacteria colonizing the wound could be detected after treatment with the maggot extract. After only two weeks of treatment, for example, in patient 1, *A. xylosoxidans* and *E. faecalis* could be found neither at the edge of the wound nor in the middle of the wound. This condition persisted until the last sampling after six weeks. Interestingly, no significant differences were found between the bacterial colonization of the wound margin and the wound center (Table 2). The initial *P. aeruginosa* isolate from patient 3 was found to be resistant to three out of four lead antibiotics (acylureidopenicillins, third- and fourth-generation cephalosporins, and quinolones), but was sensitive to all the carbapenems tested and was therefore classified as 3MRGNs (multidrug-resistant Gram-negative rods). The treatment of this patient with maggot extract was discontinued during week 3 in favor of antibiotic therapy, so no further samples could be taken. In the fourth patient at the beginning of treatment, the treated wound was colonized with *P. aeruginosa*. Owing to an increase in colonization, systemic antibiosis was initiated. In all the other patients, a decrease in the wound colonization by *S. aureus* was detected. The decrease in colonization could also be correlated with improved clinical findings. A detailed overview of wound colonization during therapy is presented in Table 2.

### 3.2. Effects of Maggot Extracts on Bacterial Growth In Vitro

The studies on growth inhibition, in the form of a zone of inhibition assay, showed no inhibition with the different extract batches or the native ESs, either in the isolated strains or in the reference strains. In the microbroth dilution assays with the patient isolates and ATCC strains for the first group of bacteria (*P. mirabilis* isolates and an ATCC strain, *P. rettgeri*, *E. faecalis*, *A. xylosoxidans*, *S. pyogenes* ATCC, and a clinical isolate of *K. oxytoca*), no significant growth inhibition was found, even at the highest possible/solvable extract concentration. The same applied to the other isolates and strains tested (*P. aeruginosa* from the wounds of patients 3, 6, and 8; *S. aureus* from the wounds of patients 1, 5, and 7; and one MRSA ATCC strain). In some isolates, even a slight but not significant increase in growth compared to the control experiments could be observed (Figure 3 and Figure 4). Total growth inhibition only occurred, according to specific antibiograms, at high concentrations of piperacillin/tazobactam. All three patient isolates of *S. aureus* were sensitive to 20/2.5 μg/mL and, in part also, to 2/0.25 μg/mL piperacillin/tazobactam, whereas the MRSA strain as expected did not respond to these concentrations with complete growth inhibition.

### 3.3. Effects of Lyophilized Maggot Extracts on Bacterial Biofilm Formation In Vitro

The investigation into the potential influence of the maggot extract on the formation of bacterial biofilms demonstrated for the first set of bacteria (*P. mirabilis* isolates and ATCC strain, *P. rettgeri*, *E. faecalis*, *A. xylosoxidans*, *S. pyogenes* ATCC, and a clinical *K. oxytoca* isolate), that for all *P. mirabilis* isolates/strain, *E. faecalis*, *S. pyogenes,* and *K. oxytoca*, the formation of biofilms in the presence of the maggot extract was inhibited. In contrast, bacterial growth and biofilm formation were completely inhibited by piperacillin/tazobactam concentrations of 2/0.25 μg/mL. For *A. xylosoxidans*, biofilm formation could only be prevented by a piperacillin/tazobactam concentration of 20/2.5 μg/mL. The lower piperacillin/tazobactam concentration (2/0.25 μg/mL) or the maggot extract resulted in an approximately equally strong biofilm formation compared to that of the control (Figure 5).

In the second set of bacteria tested (*P. aeruginosa* from the wounds of patients 3, 6, and 8; *S. aureus* from wounds of patients 1, 5, and 7; and MRSA and MSSA ATCC strains), the maggot extract resulted in a slight increase in biofilm formation in one *S. aureus* isolate (Figure 6). A minimal or no inhibitory effect on biofilm formation was observed for the *P. aeruginosa* isolate from patient 3 and the MSSA strain, with a comparable effect of piperacillin/tazobactam 2/0.25 μg/mL on the *P. aeruginosa* from patient 3. However, a higher concentration of 20/2.5 μg/mL piperacillin/tazobactam resulted in the complete prevention of biofilm formation in both strains. The extract demonstrated a significant inhibitory effect on biofilm formation by the *P. aeruginosa* isolates from patient 6, as well as the *S. aureus* isolate of patient 1, and an almost complete inhibitory effect on biofilm formation by the *P. aeruginosa* isolate from patient 8 and the *S. aureus* isolate from patient 7, comparable to that of piperacillin/tazobactam at a concentration of 2/0.25 μg/mL. For the strong biofilm-forming MRSA strain, biofilm formation was almost completely prevented with the maggot extract, comparable to that of high-dose piperacillin/tazobactam. Lower doses of piperacillin/tazobactam resulted in enhanced biofilm formation.

### 3.4. Clinical Course of Lower Leg Ulcerations During Treatment with Lyophilized Maggot Extract

This proof-of-principle study included 10 patients with chronic wounds that had undergone treatment. The dimensions, duration, and etiologies of the underlying ulcerations were correspondingly heterogeneous. Rigorous statistical analyses cannot be conducted due to the limited sample size and the heterogeneous patient population (Table 2). In two patients, the treatment regimens were discontinued after two weeks due to an inadequate therapeutic response or due to the increasing colonization with *P. aeruginosa*, necessitating antibiotic therapy. Another patient (patient 9) was transitioned to an alternative therapeutic approach after approximately seven weeks due to critical colonization with *P. aeruginosa*. In the remaining seven patients, a significant reduction in fibrin deposits and an increase in granulation tissue were observed after only two weeks. Nearly complete wound closure was achieved in patients 7 and 8 by the conclusion of the treatment period after 8 weeks. The ulcerations of patients 1, 2, 5, 6, and 10 exhibited signs of secondary wound healing, including epithelial islets, re-epithelialization of the wound edges, fibrin depletion, and consecutive increases in the granulation tissue, which were not attained via treatment prior to this investigation (Table 3 and Figure 7).

## 4. Discussion

Traditional maggot therapy has recently gained significance for treating persistent wounds that are challenging to heal and have been fully treated [20]. Our study also focused on this patient group, as they present significant challenges for physicians in their daily clinical practice. In a systematic comparison of conventional treatment methods for chronic ulcerations, it was found that maggot therapy significantly accelerates the healing of wounds, particularly diabetic foot syndrome, but also of other ulcers, and increases the likelihood of healing [21]. A recent study by Wu et al. showcased the efficacy of maggot extract at enhancing diabetic rat skin wound healing. This was achieved through the stimulation of STAT3 signaling activity and the increased expression of STAT3’s downstream target genes, including Bcl-2, Cyclin D1, and VEGF. Given these properties, maggot extract has presented itself as a promising solution for addressing the intricate process of skin wound healing in diabetic individuals [16]. The growing significance of biofilm formation and its associated chronic inflammation for perpetuating wound healing disorders has become increasingly evident, alongside the treatment of the underlying causes of chronic ulcerations [22,23]. Chronic wound infections are characterized by ongoing biofilm-related inflammation and damage to the host tissues, displaying clinical indicators that differ from those of acute wound infections. The biofilm in chronic cases also demonstrates increased resistance to antimicrobial treatments compared to acute infections. The current approaches to wound healing stress the significance of early anti-biofilm interventions to prevent biofilm formation from hindering the healing process [22].

Given this context, the objective of this study was to investigate the possibility of making the beneficial effects of this biosurgical debridement accessible to a patient population by circumventing the adverse effects of maggot therapy. Notably, biosurgical debridement is frequently viewed as a treatment of last resort, suggesting the existence of potential systemic and individual obstacles to its use that are not present with other debridement methods. An initial review of the literature revealed three primary themes or barriers to the implementation and use of this debridement method. These encompassed possible negative attitudes and stigma among healthcare providers and patients; issues related to the expense, training, and availability of medicinal maggots; and finally, specific adverse effects or contraindications associated with the use of live larvae [13].

The extracts from the larvae of *Lucilia sericata* significantly differ from the method of biosurgical debridement with the living larvae of this species. Locally administered maggots continuously release their excretions/secretions into a wound and thus to the bacteria present there. It has been established that the larvae of *Lucilia sericata* possess various proteases and antimicrobial peptides (AMPs), such as seraticin, and evidence from in vitro experiments has indicated that living maggots are capable of affecting their bacterial environment by expressing certain antibacterial peptides, such as putative Lucilia defensin, diptericin, and various proline-rich antibacterial peptides [24,25,26,27,28].

The absence of a bactericidal effect in this maggot extract compared to living fly maggots is likely attributable to the production-related denaturation/absence of these peptides and proteases. The examination of the smears of the wounds treated with the maggot extract revealed a uniform distribution of the bacterial colonization between the wound edge and center. The number of patients treated in this study (n = 10) was relatively small, but demonstrated similarities regarding the isolated species with a retrospective ten-year study conducted in Germany. The study by Jockenhofer et al., which examined the wounds of 100 patients, identified *S. aureus* (53%) and *P. aeruginosa* (25%) as the most common wound-colonizing species [29]. In the present study, *S. aureus* was found in six of the ten patients and *P. aeruginosa* in eight of the ten patients. Notably, the *P. aeruginosa* isolates exhibited a remarkable intrinsic antibiotic resistance in this study, with one isolate being multidrug resistant to three out of four antibiotic classes (3MRGNs), which could render colonization particularly critical.

During the course of this study, it was observed that the patients with a short ulcer persistence (approximately 3 months) exhibited no colonization with *P. aeruginosa*. The ulcer with the shortest duration colonized by *P. aeruginosa* had existed for six months (patient 3). Given the relatively short six-month duration of the ulcer, along with the increasing green plaque formation and the detection of *P. aeruginosa*, this colonization could be postulated as the cause of the ulcer’s persistent nature. A longitudinal study comparable to this investigation demonstrated that the duration of an ulcer increases the likelihood of colonization by *P. aeruginosa*, and this colonization can result in ulcer enlargement and further delay wound healing [30]. This phenomenon was also clinically evident in the three patients who had to prematurely discontinue this study due to increased local colonization by *P. aeruginosa*. Nevertheless, with the consistent application of the maggot extract in a total of seven patients, a clear clinical improvement was observed; therefore, it can be inferred that its positive effects on wound healing are not primarily mediated by the direct antimicrobial effect of the extract. Our analyses also revealed that the extract exhibited a remarkable inhibitory effect on the formation of bacterial biofilms. These biofilms play an essential role in the impaired healing of chronic wounds, as they lead to prolonged inflammation and concomitant delayed wound healing by inhibiting local immune reactions [31]. This result is supported by the results of Becerikli et al. who, in experiments with the same larval extract, found that the stability of the biofilms of *P. aeruginosa* and *S. aureus* was significantly impaired, especially in the presence of antibiotics [17].

It is important to note that improvement in wound healing is not necessarily measured solely by a reduction in the ulcer area. Additional factors, such as the formation of granulation tissue at the wound base and a consequent reduction in wound depth, as well as the status of the proteinases and growth factors present in the wound exudates, reflect alterations in the wound status. For instance, a decrease in biofilm could lead to a positive change in the wound environment with regard to improved wound healing [22,32]. Specifically, the colonization of biofilm-forming species, some of which exhibit pronounced resistance to antibiotics, are also likely to negatively influence wound healing through various virulence factors. These biofilms, consisting primarily of extracellular polymeric substances, provide bacteria with protection from external influences, such as antibiotics [33]. The dissolution of such biofilms or the prevention of their formation was demonstrated for almost all the isolated species in this study (Figure 5 and Figure 6); therefore, it appears to be of essential importance for the healing of chronic wounds. Consequently, it is highly probable that the improved clinical situation of the chronic wounds of the patients in this study is attributable to the reduction in biofilm formation, leading to the enhanced accessibility of the host’s immune system and the reduced proliferation of colonizing bacteria.

## 5. Conclusions

In recent years, maggot debridement therapy has gained significance for the treatment of chronic wounds. This method demonstrates both efficacy and safety; however, its widespread adoption necessitates a comprehensive knowledge among healthcare professionals and education for patients and their caregivers to foster a positive perception. Nevertheless, the application of maggot therapy presents certain challenges for patients, thereby limiting its utilization. Given the paucity of studies examining the impact of maggot extract on wound healing, we sought to investigate its effects on patients with recalcitrant chronic wounds. Our research is the first to demonstrate that *Lucilia sericata* extract has a sustained positive impact on wound cleansing and conditioning. This extract can be applied in both outpatient and inpatient settings during any dressing change, without temporal constraints, and can be readily administered by non-medical or nursing personnel following a brief instruction or the by the patient themself. Based on our observations, we anticipate that this extract will augment the treatment options for chronic wounds, offering an alternative to traditional biosurgery. We postulate that the primary indication for our lyophilized maggot extract will be as an adjunct to conventional enzymatic or autolytic wound debridement products. This study serves as an initial demonstration of concept, involving a limited number of cases in a challenging patient group. Our future plans include conducting more extensive randomized, controlled trials focusing on a specific patient population with common chronic wound healing conditions, such as persistent diabetic foot ulcers and venous leg ulcers.

## Figures and Tables

**Figure 1 biomedicines-13-00582-f001:**
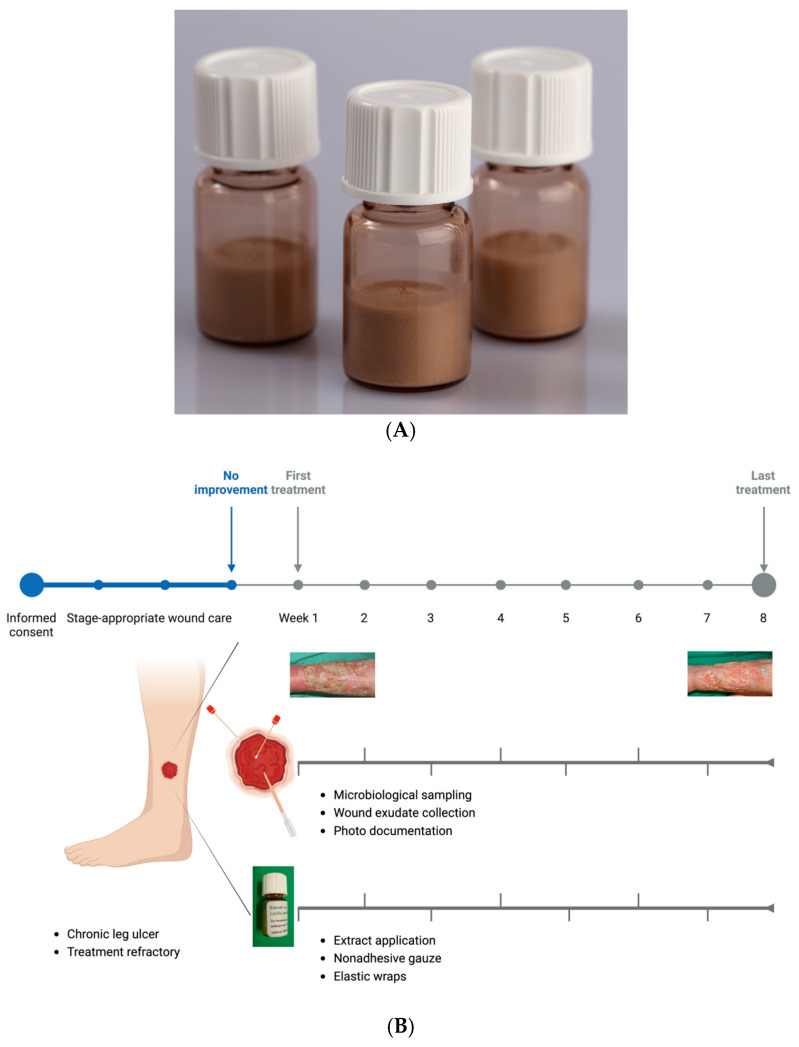
(**A**) Lyophilized extract of larvae of *Lucilia sericata* in 3.5 mL glass vials. (**B**) Procedure for the lyophilized maggot extract studied. Patients with chronic wounds persisting for ≥3 months underwent a three-week period of stage-appropriate wound management without surgical intervention. If no improvement in the wound condition was observed during this time, the patient was included in this study. Microbiological swab samples were collected from both the wound edge and center at predetermined intervals. The wound exudate was obtained using a pipette, and photo documentation was performed alongside the application of lyophilized maggot extract. Created in BioRender. Homey, B. (2025) (https://BioRender.com/i00k077, accessed on 30 December 2024).

**Figure 2 biomedicines-13-00582-f002:**
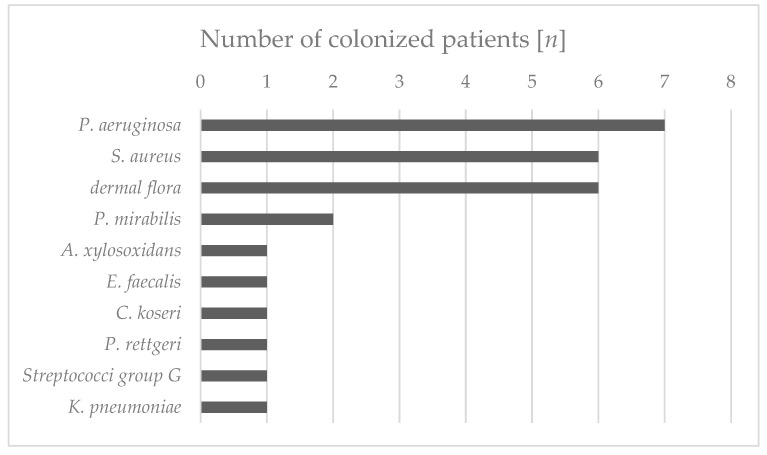
Frequencies of different bacterial species in patient wounds during the treatment with maggot extract. Swabs were taken from the wound edges and wound centers of the patients’ wounds prior to treatment and every other week of treatment. The characterization as an individual isolate was carried out via antibiograms.

**Figure 3 biomedicines-13-00582-f003:**
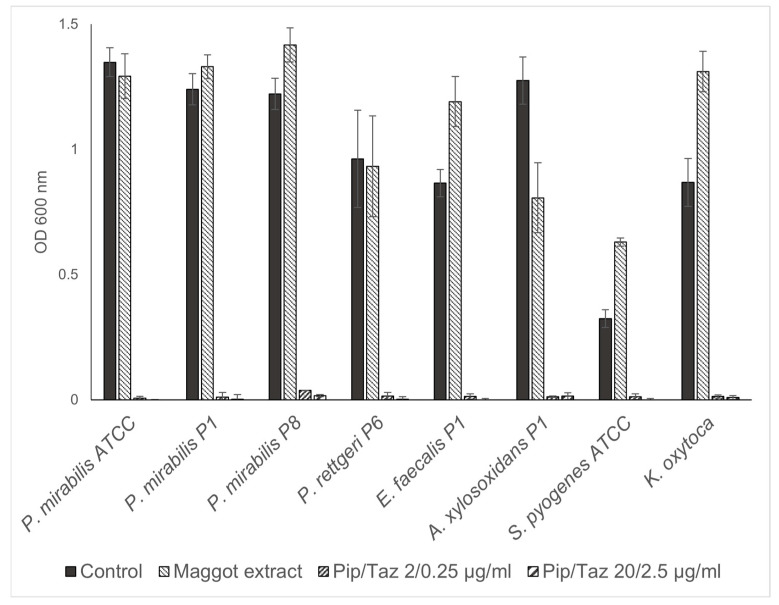
Microbroth dilution assay for *P. mirabilis* and other isolates. Isolates were cultivated in TSB (control), maggot extract dissolved in TSB, or pip/taz diluted in TSB for 24 h at 37 °C. Shown are mean values minus non-inoculated vehicle controls and standard deviation from each n = 3 measurements. OD = optical density; P = patient + number; Pip/Taz = piperacillin/tazobactam; TSB = tryptic soy broth.

**Figure 4 biomedicines-13-00582-f004:**
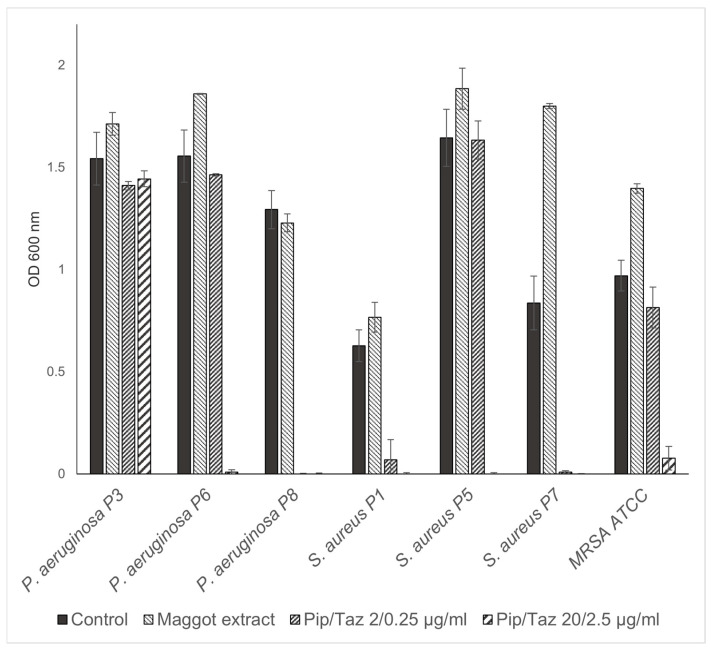
Microbroth dilution assay for *P. aeruginosa* and *S. aureus* isolates. Isolates and ATCC strains were cultivated in TSB (control), maggot extract dissolved in TSB, or pip/taz diluted in TSB for 24 h at 37 °C. Shown are mean values minus non-inoculated vehicle controls and standard deviation from each n = 3 measurements. OD = optical density; P = patient + number; Pip/Taz = piperacillin/tazobactam; TSB = tryptic soy broth.

**Figure 5 biomedicines-13-00582-f005:**
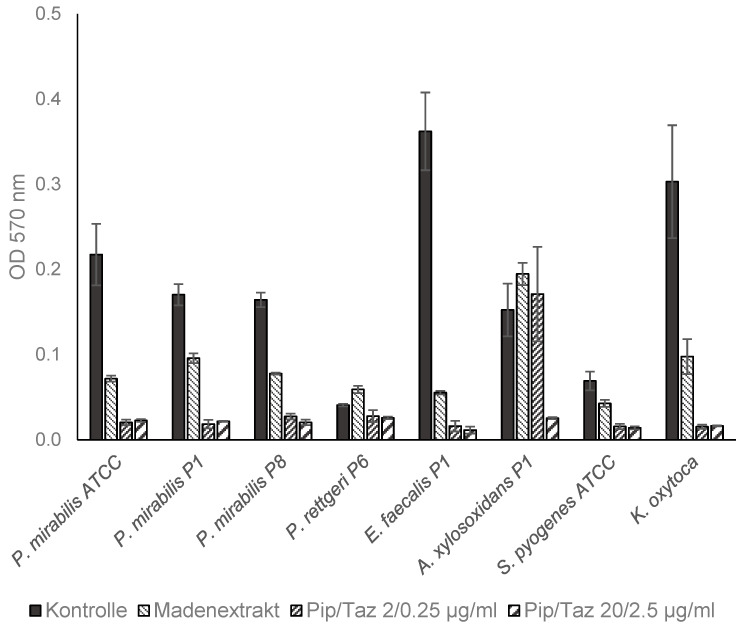
Biofilm formation by *P. mirabilis* and other isolates. Isolates and ATCC strains were cultivated in TSB (control), maggot extract dissolved in biofilm–medium, or pip/taz diluted in biofilm–medium for 24 h at 37 °C. Biofilms were stained with crystal violet. Shown are mean values minus non-inoculated vehicle controls and standard deviation from each n = 3 measurements. OD = optical density; P = patient + number; Pip/Taz = piperacillin/tazobactam.

**Figure 6 biomedicines-13-00582-f006:**
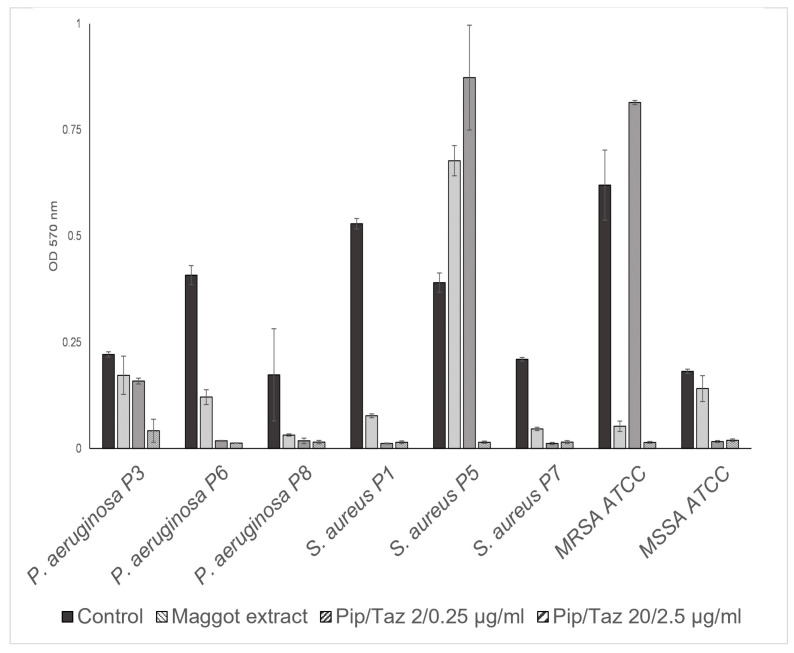
Biofilm formation by *P. aeruginosa* and *S. aureus*. Isolates and ATCC strains were cultivated in TSB (control), maggot extract dissolved in biofilm–medium, or pip/taz diluted in biofilm–medium for 24 h at 37 °C. Biofilms were stained with crystal violet. Shown are mean values minus non-inoculated vehicle controls and standard deviation from each n = 3 measurements. OD = optical density; P = patient + number; Pip/Taz = piperacillin/tazobactam.

**Figure 7 biomedicines-13-00582-f007:**
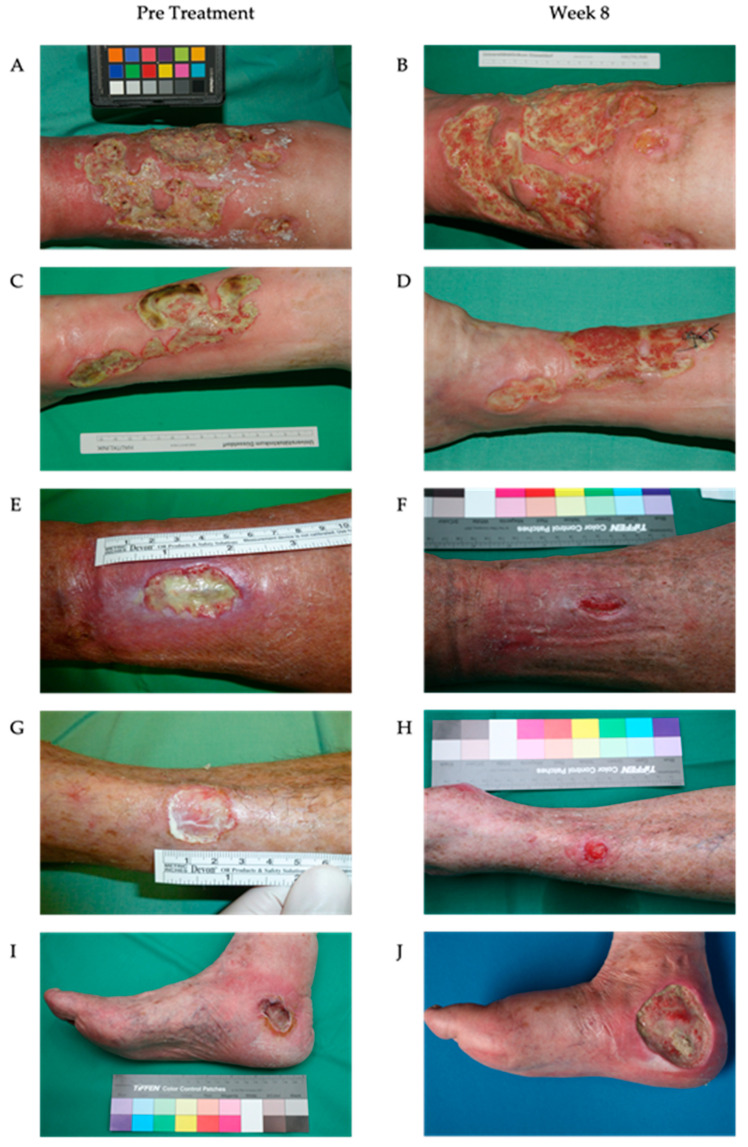
The wounds of the patients during the treatment with maggot extract. The wounds of the patients were treated three times a week. Photographs before treatment (**A**,**C**,**E**,**G**,**I**) and after eight weeks (**B**,**D**,**F**,**H**) of treatment. The treatment with the lyophilized maggot extract significantly reduced fibrin deposits and enhanced granulation tissue formation (**B**,**D**,**F**,**H**). In patients 7 (−94% wound size reduction) and 8 (−93% wound size reduction) (**E**,**F**,**G**,**H**), a marked reduction in the wound size was observed after 8 weeks, with near-complete healing and epithelialization progressing from the wound margins (**F**,**H**). In contrast, patient 9 (+471% wound size increase) (**I**,**J**) experienced a critical colonization by *Pseudomonas aeruginosa*, which was associated with an increase in the wound size.

**Table 1 biomedicines-13-00582-t001:** Overview of the treated patients. The table shows an overview of age, sex, type of wounds and their duration, and duration of treatment with lyophilized maggot extract. ♀: female; ♂: male. Average age of the patients: ♀ = 66 years; ♂ = 74 years; ♀ + ♂ = 70 years.

Patient	Sex	Age (Years)	Wound Type	Wound Duration	Treatment Duration (Weeks)	Treatment
1	♀	46	Venous	12 months	8	Clinic
2	♀	72	Venous	12 months	7	Clinic
3	♀	74	Venous	6 months	2	Clinic
4	♂	84	Venous	12 months	2	Clinic
5	♂	78	Arterial and venous	18 months	8	Patient/clinic
6	♂	65	Venous	24 months	8	Patient/clinic
7	♀	61	Postoperative	3 months	8	Clinic
8	♀	79	Venous	36 months	8	Clinic
9	♂	73	Arterial	3 months	7	Patient/clinic
10	♂	72	Venous	24 months	8	Patient/clinic

**Table 2 biomedicines-13-00582-t002:** Frequency and localization of isolation of bacterial species during treatment with maggot extract.

	Week 0	*n* = 10	Week 2	*n* = 9	Week 4	*n* = 8	Week 6	*n* = 8	Week 8	*n* = 5
	margin	center	margin	center	margin	center	margin	center	margin	center
*P. aeruginosa*	7	6	6	5	6	4	3	4	3	3
*S. aureus*	6	5	2	2	2	2	4	4	1	1
*P. mirabilis*	2	1	1	1	0	0	2	1	1	1
*A. xylosoxidans*	1	1	0	0	0	0	0	0	0	0
*E. faecalis*	1	1	0	0	0	0	0	0	0	0
*P. rettgeri*	1	0	0	0	0	0	0	0	0	0
*C. koseri*	1	1	1	1	1	1	0	0	0	0
*K. pneumoniae*	0	0	0	0	0	0	1	1	0	0
*Gr. G Streptococci*	1	0	1	0	1	0	1	0	1	0
*Dermal flora*	4	3	3	2	0	0	2	2	2	2

**Table 3 biomedicines-13-00582-t003:** Wound areas were measured following the initial run-in period at the commencement of this study and subsequently at the conclusion of this study (EOT), with the measurements recorded in square centimeters. The observed changes in the wound area were categorized as either a reduction (green) or no reduction/increase (red). The presence or absence of granulation tissue was evaluated by three independent wound specialists at the clinical facility, with assessments recorded as either present or absent. EOT = end of treatment. ♀: female; ♂: male.

Patient	Sex	Age (Years)	Wound Size Before Study (cm^2^)	Wound Size EOT (cm^2^)	Wound Size EOT (%)	Granulation Tissue Absent/Present
1	♀	46	288	202.5	−30	present
2	♀	72	66	42.75	−33	present
3	♀	74	6.25	6.25	0	absent
4	♂	84	143	143	0	absent
5	♂	78	12	6.75	−44	present
6	♂	65	90	65	−28	present
7	♀	61	8.75	0.5	−94	present
8	♀	79	7.5	0.48	−93	present
9	♂	73	8.75	41.25	+471	absent
10	♂	72	19.25	12	−38	present

## Data Availability

The raw data supporting the conclusions of this article will be made available by the authors on request.

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
