# Peer review of "Lyophilized Extract from the Larvae of the Blowfly Lucilia sericata as a New Strategy for the Management of Chronic Wounds"

_biomedicines, 2025, doi:10.3390/biomedicines13030582_

Round 1
Reviewer 1 Report
Comments and Suggestions for Authors
The study developed and tested a lyophilized extract of Lucilia sericata larvae, demonstrating its potential to reduce bacterial biofilms and enhance the healing of chronic wounds. While the study is well-designed, improvements are needed to enhance the quality of the manuscript.
- In the "Methods" section of the abstract, it would be helpful to include a brief explanation of the methods used to evaluate the efficacy of the prepared lyophilized extract in promoting the wound healing process.
- It would be helpful to include a brief background on the wound healing process in the "Introduction" section. (For the reference: https://doi.org/10.1093/burnst/tkae069 and https://www.mdpi.com/1422-0067/25/7/3790)
- In subsection "2.5. Bacterial Colonization Analysis," it would be helpful to provide a brief explanation of the bacterial culture method used.
- Why didn’t the authors perform any statistical analysis for this study? Incorporating statistical analysis could have strengthened the reliability of the findings, provided quantitative validation, and enhanced the overall scientific rigor of the study.
- The study would benefit from incorporating more recent references to provide background information and contextualize the obtained results. Updating the references will ensure the research is aligned with the current state of the field.
- The "Conclusions" section could be improved. Firstly, it’s important to clearly state the research question or objective to remind readers of the study’s purpose. Next, the key findings should be succinctly summarized and interpreted, highlighting their implications and contributions to existing knowledge in the field. Additionally, suggesting future research directions based on these findings would be valuable, identifying areas in need of further investigation.
Author Response
|
Summary
|
|
|
|
Thank you very much for taking the time to review this manuscript. Please find the detailed responses below and the corresponding revisions in track changes in the re-submitted files.
|
||
|
Point-by-point response to Comments and Suggestions for Authors
|
||
|
Comments 1: In the "Methods" section of the abstract, it would be helpful to include a brief explanation of the methods used to evaluate the efficacy of the prepared lyophilized extract in promoting the wound healing process.
|
||
|
Response 1: Many thanks for the important information. A corresponding phrase has been added to the methods section of the abstract. „Wound healing was documented photographically and measured visually in terms of a reduction in the wound slough, an increase in granulation tissue and a reduction in the wound area.”
|
||
|
Comments 2: It would be helpful to include a brief background on the wound healing process in the "Introduction" section. (For the reference: https://doi.org/10.1093/burnst/tkae069 and https://www.mdpi.com/1422-0067/25/7/3790)
|
||
|
Response 2: Thanks for the comment. We have added a paragraph with the cited reference in the introduction.
Comments 3: In subsection "2.5. Bacterial Colonization Analysis," it would be helpful to provide a brief explanation of the bacterial culture method used.
Response 3: We agree and have added a corresponding passage. „The swabs were automated on Columbia agar with 5 % sheep blood (bioMérieux Deutschland GmbH, Nürtingen, Germany) and MacConkey agar (Oxoid Deutschland GmbH, Wesel, Germany) and incubated at 37°C for 48 hours. The identification and resistance testing was automated using VITEK II® (bioMérieux Deutschland GmbH, Nürtingen, Germany) or by agar diffusion for pseudomonads. Resistance was assessed according to CLSI (Clinical and Laboratory Standards Institute).“
Comments 4: Why didn’t the authors perform any statistical analysis for this study? Incorporating statistical analysis could have strengthened the reliability of the findings, provided quantitative validation, and enhanced the overall scientific rigor of the study.
Response 4: We agree with the assumption that a statistical analysis leads to an increased scientific reliability of results. This study was a pilot study, which was intended to cover a certain spectrum of wounds of different genesis to be treated, with the aim of finding out whether there is a therapeutic target that can be qualitatively evaluated at all. We were therefore limited by the ethics vote to a maximum of 10 patients, which is a number too small for the usual statistic methodology to be able to test the significance further. However, on the basis of the results presented here, a study is planned on a much larger patient collective, which will allow various quantifiable parameters to be analysed statistically.
Comments 5: The study would benefit from incorporating more recent references to provide background information and contextualize the obtained results. Updating the references will ensure the research is aligned with the current state of the field.
Response 5: We appreciate your comment. We have examined the existing research on chronic wounds, maggot therapy, and maggot extract therapy. As a result, we have incorporated relevant papers that are significant to our study and expanded the discussion section accordingly.
Comments 6: The "Conclusions" section could be improved. Firstly, it’s important to clearly state the research question or objective to remind readers of the study’s purpose. Next, the key findings should be succinctly summarized and interpreted, highlighting their implications and contributions to existing knowledge in the field. Additionally, suggesting future research directions based on these findings would be valuable, identifying areas in need of further investigation.
Response 6: We appreciate your valuable feedback. In response to your recommendations, we have updated our conclusions to reflect the current research landscape, our findings, and potential areas for future investigation.
|
||

Reviewer 2 Report
Comments and Suggestions for Authors
You state: 'a marked reduction in wound size was observed after 8 weeks'.
Please could you back this up with the data from the trial, for example percentage area reduction.
Alternatively remove this and all similar statements.
Author Response
|
Summary
|
|
|
Thank you very much for taking the time to review this manuscript. Please find the detailed responses below and the corresponding revisions in track changes in the re-submitted files.
|
|
|
Point-by-point response to Comments and Suggestions for Authors
|
|
|
Comments 1: You state: 'a marked reduction in wound size was observed after 8 weeks'. Please could you back this up with the data from the trial, for example percentage area reduction. Alternatively remove this and all similar statements
|
|
|
Response 1: We appreciate your valuable comment. We have incorporated your recommendations and, in addition to the purely clinical description of the wounds and their size, we have included an analysis of the area reduction in the text and in table 3. |
|

Reviewer 3 Report
Comments and Suggestions for Authors
In this article, authors have used lyophilized extract of larvae of Lucilia sericata, as an alternative to use of live larvae for debridement of wounds. It is a good idea to use and explore alternative ways to treat chronic wounds with good patient compliance, although here number of patients utilized are low. I recommended this article for publication with following changes:
· Introduction lacks data on use of such extract previously for e.g. Life 2022, 12(2), 237; https://doi.org/10.3390/life12020237 . Please add more references.
· Results of test mentioned in section 2.1, such as LAL test, skin irritancy, toxicity testing etc. are missing. Please add the results in the text.
· In section 2.1, author used water to extract the larvae, any particular reason for using it. Secondly, how to obtain the extract emulsion? Did authors added any excipients?
· What about the standardization of extract and quantification of subcomponents of extracts? It is very important to know the exact quantity of them.
· How dose or amount of extract employed in patients was calculated? It is not clear how much quantity was there in 3.5ml vials?
· Did authors, found any difference when the vial of extract was used thrice or twice a week?
· Any suitable explanation for development of P. aeruginosa in limited patients treated with extract of larvae?
Author Response
|
Summary
|
|
|
Thank you very much for taking the time to review this manuscript. Please find the detailed responses below and the corresponding revisions in track changes in the re-submitted files.
|
|
|
Point-by-point response to Comments and Suggestions for Authors
|
|
|
Comments 1: Introduction lacks data on use of such extract previously for e.g. Life 2022, 12(2), 237; https://doi.org/10.3390/life12020237 . Please add more references.
|
|
|
Response 1: We appreciate your valuable input. In response, we have conducted a thorough examination of existing research on chronic wounds, maggot therapy, and maggot extract therapy. We have incorporated relevant and significant studies related to our field of research into the introductory section, providing a comprehensive context for our work.
|
|
|
Comments 2: Results of test mentioned in section 2.1, such as LAL test, skin irritancy, toxicity testing etc. are missing. Please add the results in the text.
|
|
|
Response 2: The results described here are part of the risk assessment/biological evaluation in accordance with ISO 10993, which were carried out in various external laboratories and were necessary for entry into clinical testing and would go far beyond the scope of this article. We will therefore delete the passage "The safety of the extract was evaluated using the HET-CAM test, dermatological skin tests, pyrogen tests (rabbit test, LAL assay, and monocyte activation test), and in vitro toxicity assays, all of which indicated an absence of toxic or allergic reactions." from 2.1.
Comments 3: In section 2.1, author used water to extract the larvae, any particular reason for using it. Secondly, how to obtain the extract emulsion? Did authors added any excipients?
Response 3: The emulsion was produced using a knife mill; this information has been added to the manuscript. As stated in the text, the extraction is carried out exclusively with water as excipient, as this can be removed almost residue-free by subsequent lyophilisation and does not represent any toxic/irritating potential and in order to produce an extract suitable for storage. The water-insoluble components were removed during sedimentation/filtration. The exact production method is the intellectual property of Alpha-Biocare GmbH and may not be published in detail. Essentially, it is intended to express that a heated, aqueous/water-soluble, sterile extract can be produced in the form of a powder.
Comments 4: What about the standardization of extract and quantification of subcomponents of extracts? It is very important to know the exact quantity of them.
Response 4: In fact, this is a standardised method. Unfortunately, we have to disappoint you here too, as the exact parameters of the extraction and the detailed composition are proprietary knowledge of Alpha-Biocare GmbH. However, with the information provided here, it should also be possible to produce a comparable extract, should anyone wish to do so.
Comments 5: How dose or amount of extract employed in patients was calculated? It is not clear how much quantity was there in 3.5ml vials?
Response 5: You are right, this information is missing. 2 ml extract was filled into the vials and freeze-dried and also resuspended with 2 ml saline. The relevant information has been added to the text. The required/applicable amount was determined empirically by the treating physicians in this study.
Comments 6: Did authors, found any difference when the vial of extract was used thrice or twice a week?
Response 6: This addresses a significant question regarding the future frequency of use of the extract. In this proof-of-principle study, no definitive conclusions could be drawn about the differences and effectiveness of the frequency of use due to the limited sample size. This limitation represents a crucial point that should be addressed in subsequent research. The input provided is appreciated.
Comments 7: Did authors, found any difference when the vial of extract was used thrice or twice a week?
Response 7: This is an interesting question, which we have also asked ourselves, but which we cannot conclusively explain due to the natural diversity/complexity of the different P. aeuriginosa strains. Our strong suspicion is that certain strains can be forced into a planktonic status by this extract from the organisation in biofilms. This could then lead to increased invasiveness compared to organisation in biofilms, but also to better accessibility for antibiosis, which is otherwise difficult to achieve. However, this topic is very extensive due to the large number of variables and could not be clarified within the scope of this study. |
|

Reviewer 4 Report
Comments and Suggestions for Authors
Manuscript ID: biomedicines-3429086 "Lyophilized extract from larvae of the blowfly Lucilia sericata as a new strategy for the management of chronic wounds"
Overall Impression
This manuscript addresses an important and innovative area of research: the use of a lyophilized extract from Lucilia sericata larvae for managing chronic wounds. While the topic is highly relevant and shows promise, certain aspects of the manuscript require significant improvement to enhance clarity, rigor, and presentation.
Major Comments
Abstract:
1. The abstract effectively summarizes the background and objectives, but it could be strengthened by incorporating specific numerical data (e.g., the percentage of wound healing or reduction in bacterial colonization) and emphasizing the novelty of the approach.
2. Clearly state how the findings compare to existing treatments and their clinical implications.
Introduction:
3. The introduction provides sufficient background on chronic wounds and the limitations of conventional treatments, but it does not adequately explain the rationale behind the development of the lyophilized extract. A clearer justification for why this approach was chosen over others is needed.
4. Include more recent references on biosurgical debridement and antimicrobial peptides derived from Lucilia sericata to contextualize the innovation.
Materials and Methods:
5. The methodology for preparing the lyophilized extract is detailed; however, the steps for ensuring batch-to-batch consistency should be clarified. This is crucial for reproducibility.
6. Ethical considerations are mentioned, but further details about patient recruitment and exclusion criteria would strengthen this section.
7. Expand on the rationale for the assays used (e.g., the choice of bacterial strains and their relevance to chronic wounds).
8. For the biofilm and bacterial growth assays, specify the controls used in more detail. For instance, were there untreated wounds or wounds treated with standard care for comparison?
Results:
The results are well-organized, but some key findings are not sufficiently emphasized. For instance:
9. Highlight the significance of the reduction in biofilm formation and bacterial colonization.
10. Discuss why P. aeruginosa colonization persisted in some patients and how this influenced outcomes.
Figures and tables are generally clear, but:
11. Figures 9 and 10 require clearer labeling of lines and legends for ease of interpretation.
12. Figures 5 and 6 could benefit from additional annotations to highlight significant differences.
13. Include statistical analyses for results presented (e.g., bacterial reduction rates).
Discussion:
The discussion lacks depth in linking the results to the broader clinical context. Specifically:
14. Elaborate on how the findings contribute to advancing the field of wound management.
15. Address potential limitations of the study, such as the small sample size and lack of a control group.
16. Discuss the potential mechanisms through which the extract reduces biofilm formation. Is this due to retained enzymatic activity, or are other factors involved?
17. Suggest how the product could be scaled for broader clinical use, considering production and storage challenges.
Conclusion:
18. The conclusion is concise but could be expanded to include specific next steps for research, such as conducting randomized controlled trials or exploring additional bacterial strains.
19. Mention the potential limitations of the current study (e.g., small sample size) and propose solutions for future studies.
Figures and Tables:
20. Several figures could be merged for better clarity and space efficiency (e.g., Figures 1A/1B, Figures 5/6).
21. Table 2 presents valuable data but should be referenced more explicitly in the results section to highlight trends.
Improve figure resolution and ensure consistent formatting across all visuals.
Minor Comments
Grammar and Language:
22. Minor grammatical errors and typos should be corrected throughout the manuscript (e.g., "biosurgery" is occasionally inconsistently referenced).
23. Simplify technical jargon where possible to ensure accessibility to a broader audience.
24. Some abbreviations, such as MDPI, are well-known. Please consider removing them and keeping only the significant abbreviations relevant to the study.
Consistency in Terminology:
25. Ensure consistent use of terms such as "lyophilized extract," "maggot extract," and "Larveel®." 26. Avoid using multiple terms interchangeably without clarification.
Formatting:
27. Follow the journal’s guidelines for formatting references, figures, and tables to ensure consistency.
Recommendations
While this manuscript presents valuable research, improvements in clarity, methodology, and discussion are needed to maximize its impact. The innovative use of a lyophilized extract for chronic wound management has the potential to address a critical clinical need, and the authors should focus on refining the presentation to emphasize its significance.
Comments on the Quality of English Language
The English could be improved to more clearly express the research.
Author Response
|
Summary
|
|
|
|
Thank you very much for taking the time to review this manuscript. Please find the detailed responses below and the corresponding revisions in track changes in the re-submitted files.
|
||
|
Point-by-point response to Comments and Suggestions for Authors
|
||
|
Comments 1: The abstract effectively summarizes the background and objectives, but it could be strengthened by incorporating specific numerical data (e.g., the percentage of wound healing or reduction in bacterial colonization) and emphasizing the novelty of the approach |
||
|
Response 1: Thank you very much for this line of thought, which we also pursued following the study and would have liked to have implemented in this way. Unfortunately, this study was a pilot study, which was intended to cover a certain spectrum of wounds of different genesis to be treated, with the aim of finding out whether there is a therapeutic target that can be qualitatively evaluated at all. We were therefore limited by the ethics vote to a maximum of 10 patients, which is a number too small for the usual statistic methodology to be able to test the significance further. However, on the basis of the results presented here, a study is planned on a much larger patient collective, which will allow various quantifiable parameters to be analysed statistically. With regard to bacterial colonisation in particular, we expected a different picture with regard to the eradication of bacteria from the wound. It was only in the course of the study that we learned that the bacteria are not ‘killed’ by the extract, but that their organisational forms, e.g. in biofilms, may be influenced.
|
||
|
Comments 2: Clearly state how the findings compare to existing treatments and their clinical implications. |
||
|
Response 2: We believe that we have compared as much as the data situation allows. In order to make a direct comparison with a treatment method that is comparable in its basic features, it would have to be tested directly against it, which was not permitted due to the nature of the study. However, this approach is planned for further studies.
Comments 3: The introduction provides sufficient background on chronic wounds and the limitations of conventional treatments, but it does not adequately explain the rationale behind the development of the lyophilized extract. A clearer justification for why this approach was chosen over others is needed.
Response 3: Many thanks for the suggestion. We have edited the introduction to better explain the rationale for producing a maggot extract.
Comments 4: Include more recent references on biosurgical debridement and antimicrobial peptides derived from Lucilia sericata to contextualize the innovation.
Response 4: Thanks for the comment. In response, we have conducted a thorough examination of existing research on chronic wounds, maggot therapy, and maggot extract therapy. We have incorporated relevant and significant studies related to our field of research into the introductory section, providing a comprehensive context for our work.
Comments 5: The methodology for preparing the lyophilized extract is detailed; however, the steps for ensuring batch-to-batch consistency should be clarified. This is crucial for reproducibility.
Response 5: The lyophilized extract was in fact produced using a validated method that cannot be presented in its entirety. Due to the now large amount of literature on this topic and the highly variable methods used, it was important for us to present the core features of the production process (aqueous solution, heating, sterile filtration, freeze-drying and radiation sterilization). A further breakdown would have little added value for most experimenters and institutes due to the industrial devices and large volumes/quantities used.
Comments 6: Ethical considerations are mentioned, but further details about patient recruitment and exclusion criteria would strengthen this section.
Response 6: Actually, this information is given in section 2.2: „1. Chronic leg ulcers, 2. Refractory to treatment for at least three months, 3. No underlying consumptive diseases (e.g., malignancies), 4. A three-week pretreatment phase with stage-appropriate wound care demonstrated no significant improvement.”
Comments 7: Expand on the rationale for the assays used (e.g., the choice of bacterial strains and their relevance to chronic wounds).
Response 7: Thank you, we added this information in this chapter.
Comments 8: For the biofilm and bacterial growth assays, specify the controls used in more detail. For instance, were there untreated wounds or wounds treated with standard care for comparison?
Response 8: We hope we have understood your comment correctly. We have already answered part of your comment in (7). Strictly speaking, the study was a series of hard-to-heal chronic wounds in which there was no untreated control group. However, there is a comparison to a certain extent due to the frustrating healing process beforehand. The bacterial species examined in vitro were isolates obtained from the wounds of the treated patients and additionally more or less clinically relevant laboratory strains or other clinical isolates.
Comments 9: Highlight the significance of the reduction in biofilm formation and bacterial colonization.
Response 9: We have taken this aspect into account as far as possible when revising the manuscript and have incorporated it in some places.
Comments 10: Discuss why P. aeruginosa colonization persisted in some patients and how this influenced outcomes.
Response 10: Thank you for your comment. We feel that we have done this at various points in the manuscript. The persistence results, and this is our guess, from the duration of existence/colonization of the wound and the intrinsic resistance together. In order to break this down further, more in-depth molecular biological analyses would probably be necessary for each isolate, which would go far beyond the scope of this study.
Comments 11: Figures 9 and 10 require clearer labeling of lines and legends for ease of interpretation.
Response 11: We have changed the font size/layout where possible.
Comments 12: Figures 5 and 6 could benefit from additional annotations to highlight significant differences.
Response 12: Due to the sample size and on the advice of our in-house statistician, we consider the application of further statistical methods to be inappropriate. Unfortunately, due to the high intrinsic variance, it was also not possible to summarize data from the different isolates. in the planned study, we will proceed differently and aim for an n of at least 50.
Comments 13: Include statistical analyses for results presented (e.g., bacterial reduction rates).
Response 13: In addition to our response 12, we must point out that the aim was not to dissolve existing biofilms, but to investigate whether they are formed more slowly. In our opinion, inserting a reduction rate would therefore be misleading. Moreover, ‘reversing’ the scaling and scaling to % would result in the metadata being lost that not all isolates are equally good biofilm builders.
Comments 14: Elaborate on how the findings contribute to advancing the field of wound management.
Response 14: We have revised results and conclusion and hope that this information can be seen more clearly.
Comments 15: Address potential limitations of the study, such as the small sample size and lack of a control group.
Response 15: We have revised the discussion and incorporated the required recommendations.
Comments 16: Discuss the potential mechanisms through which the extract reduces biofilm formation. Is this due to retained enzymatic activity, or are other factors involved?
Response 16: This extract has no enzymatic activity due to the way it is produced. A great deal of basic research is still needed in this complex field. If we had to make a confidential assessment, we would say that it is a low-molecular sugar compound from the larval salivary glands, in which the extracorporeally released enzymes of the larvae are otherwise embedded until activation. These molecules survive many thermal and chemical treatments and are present in large quantities in the larvae. The isolation of these individual substances, although extremely interesting, is almost impossible.
Comments 17: Suggest how the product could be scaled for broader clinical use, considering production and storage challenges.
Response 17: Although this is a straight forward thought, we think that this is somewhat out of scope of this paper. However, we know that this has already been done and a product is in the process of being authorized. The storage challenge is obsolete due to the removal of water and gamma irradiation.
Comments 18: The conclusion is concise but could be expanded to include specific next steps for research, such as conducting randomized controlled trials or exploring additional bacterial strains.
Response 18: We have revised the conclusion and incorporated the required recommendations.
Comments 19: Mention the potential limitations of the current study (e.g., small sample size) and propose solutions for future studies.
Response 19: We appreciate your feedback provided in point 19. In response, we have incorporated your suggestions and modified our conclusions to include the requested information regarding the study design and future outlook.
Comments 20: Several figures could be merged for better clarity and space efficiency (e.g., Figures 1A/1B, Figures 5/6).
Response 20: Good idea. We have implemented it like this.
Comments 21: Table 2 presents valuable data but should be referenced more explicitly in the results section to highlight trends. Improve figure resolution and ensure consistent formatting across all visuals.
Response 21: We have made various adjustments to the graphics.
Comments 22: Minor grammatical errors and typos should be corrected throughout the manuscript (e.g., "biosurgery" is occasionally inconsistently referenced).
Response 22: We agree and have revised the manuscript in line with your recommendation.
Comments 23: Simplify technical jargon where possible to ensure accessibility to a broader audience.
Response 23: We believe we have taken this into account in the revision.
Comments 24: Some abbreviations, such as MDPI, are well-known. Please consider removing them and keeping only the significant abbreviations relevant to the study.
Response 24: We agree and have revised the manuscript in line with your recommendation.
Comments 25: Ensure consistent use of terms such as "lyophilized extract," "maggot extract," and "Larveel®."
Response 25: We agree and have reviewed the manuscript in light of your recommendation.
Comments 26: Avoid using multiple terms interchangeably without clarification.
Response 26: We agree and revised the manuscript.
Comments 27: Follow the journal’s guidelines for formatting references, figures, and tables to ensure consistency.
Response 27: We prepared the manuscript according to the journal's specifications.
|
||

Round 2
Reviewer 1 Report
Comments and Suggestions for Authors
The manuscript has been properly revised. I have no further suggestions.
Reviewer 2 Report
Comments and Suggestions for Authors
the authors have made the suggested changes
Reviewer 3 Report
Comments and Suggestions for Authors
In revised version authors, have adressed most of the comments except 2, 3 and 4. Please readress them. Please donot delete the data, provide its detail, which is very important before its implementation in humans.
Reviewer 4 Report
Comments and Suggestions for Authors
Thank you for your thoughtful revisions and for addressing the previous comments in a comprehensive manner. The revised manuscript demonstrates significant improvements in clarity, methodology, and presentation. Your revisions enhance the overall quality and scientific rigor of the study, making it more suitable for publication.
The responses to reviewer comments were well-justified, and the manuscript now presents a clear, well-structured, and scientifically sound study. Based on the improvements made, I believe the article meets the journal's standards and should be accepted for publication.